# Impact of Subjective and Objective Factors on Bus Travel Intention

**DOI:** 10.3390/bs12110462

**Published:** 2022-11-19

**Authors:** Qi Chen, Yibo Yan, Xu Zhang, Jian Chen

**Affiliations:** 1College of Architecture and Urban Planning, Chongqing Jiaotong University, Chongqing 400074, China; 2College of Civil Engineering, Henan University of Technology, Zhengzhou 450001, China; 3College of Traffic & Transportation, Chongqing Jiaotong University, Chongqing 400074, China; 4Jiangsu Province Collaborative Innovation Center of Modern Urban Traffic Technologies, Nanjing 211189, China

**Keywords:** built environment, psychological factors, bus travel intention, structural equation model, intermediary variable

## Abstract

Given the lack of quantitative descriptions on the interaction between psychological factors and the built environment in existing urban bus travel behavior, this study examines the simultaneous influences of the objective-built environment and subjective psychological factors on bus travel intentions. An empirical study on the influence path of bus travel intention was conducted using structural equation modeling. Then, personal attribute factors were introduced, and a linear regression model was used to explore the influence of behavioral intentions. This study uses 410 investigated samples from the residents in Zhengzhou, China. The findings proved that psychological factors play mediating roles between the travel environment and its impact on travel behaviors and confirms the validity of the description of the measurement variable with respect to the bus travel intentions proposed in the study. We also found that the retirement factor among the personal attribute factors could significantly affect bus travel intentions, which means that the retired group prefers to use buses for traveling. This study shows innovations in catching the intermediary effect of psychological factors between the built environment and travel behavior while also quantifying the effects of both subjective and objective factors when choosing bus travel.

## 1. Introduction

After more than 20 years of rapid urbanization, China has undergone tremendous changes both in terms of the urban environment and in terms of travel habits. The explosive growth of car ownership in China since the turn of the 21st century reflects the aspirations of the increasingly affluent Chinese people for private vehicles on the one hand and the changing travel patterns and habits of Chinese people on the other. The crude urban development model of the past few decades has given too much attention to the overall shape of the city layout and private car travel at the expense of developing public transport [1]. The negative effects of traffic congestion and environmental degradation brought about by the rapid growth of private cars are becoming increasingly serious [2]. This has become a focus of concern and a pressing social problem for the government to address. With this background, the government hopes to restrict the use of private cars via traffic control policies, such as a number of restriction policies and implementing congestion charges, but the results have been less than satisfactory [3,4,5]. In terms of the process of travel generation, elements of the built environment, such as land use and the distribution of facilities, are considered by scholars to be important factors influencing people’s travel behavior. Therefore, in the context of global carbon-neutral development goals, guiding green public transport travel by optimizing and adjusting the built environment has become an international focus [6].

Distinguishable from the natural environment, the built environment is the man-made environment and usually consists of three main components: land development, transportation system, and urban design [7]. Land development refers to the different patterns of land use, and it is the spatial distribution of different land development patterns that generates people’s travel needs. The transport system determines the level of transport services that can be provided by transport facilities and is an important factor influencing people’s choice of travel mode. Urban designs focus on elements, such as spatial layouts and visual characteristics, which are directly perceived by people as an urban environment and an important expression of the quality of travel services [8]. In terms of their impact on travel behavior, built environment elements can be described by five dimensions: density, diversity, design, distance to transit, and destination accessibility [9]. From the traveler’s perspective, travel choices are first formed by perceiving multiple dimensions of the built environment’s characteristics. Based on these perceptions, a perception of the built environment is then generated; subjective judgements are ultimately made based on this perception, and a sense of traveling is formed.

For government departments to guide residents into choose bus travel by optimizing the built environment, they must take into account the perceived and psychological factors of the built environment and establish a pathway from the built environment to the subjective perceptions that people use in choosing bus travel. Based on this pathway, feedback can be provided to guide the optimization of the built environment, thus guiding people to choose bus travel more frequently. This study uses psychological factors as a mediating variable to establish the pathway of the “built environment-psychological factors-intention to travel” and to analyze the causes of bus travel intentions. The study is organized as follows: Section 2 reviews the current state of research on the built environment and transport travel; Section 3 explains the ideas, methods, and indicators of the study; Section 4 selects actual case data for analysis and presents the results; Section 5 discusses the findings and results; Section 6 summarizes the conclusions.

## 2. Literature Review

### 2.1. The Impact of the Built Environment on Traveling Behavior

The study of the influence of the built environment on travel behavior can be divided in terms of the scale of the study, and the division is made with respect to aggregate and disaggregated models. The calculation method of the aggregate model is based on the use of traffic zones (neighborhoods, streets, and cities) as the fundamental study unit to analyze the overall travel characteristics of travelers in different areas or groups [10,11,12]. For example, scholars have used built environment indicators, such as residential density, amenity distribution, and public transport availability, to establish a relationship between the built environment and private car ownership [13]. The calculation method of the disaggregate model comprises examining the link between the built environment and behavioral intentions, with the individual or household as the basic research unit. For example, scholars use discrete choice models to explore the links between built environment elements, such as facility density and land use, and the investigator and people’s behavioral intentions from an individual or household perspective [14,15].

The relationship between the built environment and travel behavior has been extensively validated by scholars as their research continues. Although there are differences in the perspectives and quantitative methods used in the study of built environments and traffic behavior, scholars have reached some common conclusions. However, the conclusions drawn from these studies describe the correlations between the built environment and traffic behavior and do not explain why and how the built environment affects traffic behavior.

### 2.2. The Impact of Psychological Factors on Travel Using Public Transport

From the perspective of theory-guided practice, scholars are really concerned with the causal relationship between the built environment and travel behavior rather than the correlation between the two. By analyzing the psychological decision-making process with respect to traffic behavior and revealing the pathways through which the built environment influences traffic behavior, it is possible to assist in making decisions on built environment optimization strategies to guide residents in choosing to use buses. Therefore, psychological approaches and methods play an important role in understanding environmental problems and finding solutions. Scholars attempted to establish a link between psychological factors and behavioral intentions to explain the factors influencing travel behavior. One of these includes the theory of planned behavior (TPB), which proposes a process of rational behavior formation and assumes that human behavior is formed through thought and planning; this has provided scholars in the field of transport with new ideas for understanding travel behavior.

Since then, scholars have explored the influence of psychological factors on personal travel intentions from different perspectives on the basis of the TPB theory. Scholars have continued to enrich and innovate calculation methods, path assumptions, and the measurement variables of the model [16,17]. Some scholars have found that psychological factors, such as travel habits and environmental awareness, are thought to influence travel behavioral intentions, and their influence path hypothesis has been confirmed [18]. In addition, by integrating different theoretical modeling frameworks, such as the TPB and the TAM (technology acceptance model), it can be used to analyze the psychology of travel decisions in specific scenarios [17]. Research on travel behavior decision-making based on a psychological theoretical framework has verified the validity of psychological factors in explaining bus travel intentions, and these studies have achieved some consensus in explaining the mechanisms that generate travel behavior. However, such studies do not establish a link between the objective environment and subjective psychology, making it difficult to generate feedback on built environment optimization and leading to difficulties in applying these findings to bus-oriented built environment optimization.

### 2.3. Influencing the Mechanisms of Travel Intentions

In order to investigate the mechanisms that shape travel behavior, scholars began to focus on the combined effects of the objective-built environment and subjective psychological factors on travel behavior. For example, scholars have attempted to collect data on people’s subjective perceptions and the built environment characteristics of the respondent’s residence by using questionnaires, and the influence of subjective and objective factors on travel behavior has been investigated using mathematical and statistical methods [19]. Such studies consider the environmental and psychological factors as the same dimensional factors that can influence people’s behavioral intentions rather than considering the environmental–psychological pathway of influence. According to the stimulus organism response (SOR) theory, the process of objective environmental influence on behavior can be understood as follows: people are stimulated by objective built environment elements; they then form a subjective perception, and they then make behavioral decisions based on this perception [20]. The “environment-perception-behavior” pathway, based on the SOR framework, has been validated in various types of behavioral studies [21,22], and the pathway also provides new ideas for exploring the mechanisms of the built environment’s influence on travel behavior.

### 2.4. Summary

Based on a review of the literature, research on the relationship between built environments and travel behavior has been universally confirmed by scholars, and the psychological processes involved in the behavioral decisions of traffic travel have been more fully explained. However, the following problems remain in current research in the travel behavior field. (1) Research on the relationship between the built environment and travel behavior makes explaining the causal relationship between them difficult. (2) Studies that consider psychological factors in the formation mechanism of traffic trips fail to establish a link between psychological factors and the built environment. (3) Studies that combine subjective and objective factors have failed to establish pathways of action between environmental and psychological factors, despite considering both subjective and objective factors.

To fill these research gaps, this study considers both the effect of objective-built environments and subjective psychological factors on bus travel intention. This takes the subjective psychological factors as intermediate variables to establish the influence path of an objective-built environment on bus travel intention. The innovations of this study are as follows: (1) Proposing that psychological factors play an intermediate role in the influence of built environments on travel intention; (2) Establishing the influence path of “built environment-psychological factors-behavioral intention”. (3) Formatting a feedback mechanism for the optimization of built environments based on the influence path of “built environment-psychological factors-travel intentions”.

## 3. Methodology

### 3.1. Research Framework

In this study, subjective psychological factors were used as intermediary variables to attempt to establish the influence path between the objective-built environment and bus travel intention. The data on psychological factors were obtained via questionnaire surveys, and the data on the built environment were extracted by using geographic information. Based on the process of “environment perception decision making”, a hypothesis of the influence path of the built environment on bus travel intentions was proposed. Then, the structural equation model (SEM) was adopted to verify the proposed hypothesis and to calculate the influence path coefficient of the latent variables. A regression model was then used to identify the effective factors influencing bus travel intentions. With the integration of influence paths and the quantitative relation, the formation mechanism of bus travel intention could be revealed. The idea and flowchart for the model’s construction are shown in Figure 1.

### 3.2. Structural Model

Numerous empirical studies have shown that the psychological factors of travelers have a significant impact on people’s behavioral intentions [23,24]. TPB theory involves the attitude (ATT), subjective norm (SN), and perceived behavioral control (PBC), which are all believed to positively contribute to behavioral intentions (BI). Attitudes (which are positive or negative evaluations of something), subjective norms (which are pressures people feel to perform a behavior and a person’s motivation to comply with those pressures), and perceptual behavioral control (which is the perceived degree of difficulty people have in performing an act) are influences that have been widely shown to affect people’s behavioral intentions [17,25,26]. Based on this, the hypothesis of H1, H2, and H3 are proposed as follows.

**Hypothesis 1** **(H1).**
*The level of acceptance for bus travel has a positive impact on bus travel intention.*


**Hypothesis 2** **(H2).**
*The level of the traveler’s perceived societal stress has a positive impact on bus travel intention.*


**Hypothesis 3** **(H3).**
*The traveler’s control over their behavior has a positive impact on bus travel intention.*


The influence of the built environment on behavioral intentions has also been widely confirmed [27,28]. The SOR theory’s hypothesis of environmental–psychological–behavioral intentions has been widely confirmed [20,21]. Scholars believe that travelers are first stimulated by the objective travel environment before forming a subjective understanding regarding bus travel intentions. Based on this understanding, along with other subjective factors, travelers develop travel intentions. The following hypotheses are thus proposed. The complete structural model is shown in Figure 2.

**Hypothesis 4** **(H4).**
*A better travel environment can affect the attitude of travelers when selecting bus travel, pushing them toward the idea.*


**Hypothesis 5** **(H5).**
*A better travel environment can increase the perception of social stress, pushing people to choose to use buses.*


**Hypothesis 6** **(H6).**
*A better travel environment has a positive impact on perceived behavioral control (i.e., travelers will be more inclined to travel by bus).*


**Hypothesis 7** **(H7).**
*A better travel environment has a positive impact on bus travel intentions.*


### 3.3. Measurement Model

The structural model in the SEM includes five latent variables, and the measurement model has 3–5 observed variables for each latent variable. All measurement variables are designed to have clear directional effects on the latent variables. Table 1 lists the description of the measurement variable.

The measurement variables of the bus travel environment are derived and identified by using insights from extant research results. Five measurement variables, which are bus-stop accessibility [1,29,30], functional diversity [31,32], road connectivity [33,34], distance to commercial center [35,36], and the distance to an urban center [37,38], were selected and described. Geographic data were obtained from the surrounding residential area. The travel environment calculation methodology is described as follows (referring to Table 2).

Bus-stop accessibility is the reflection of the convenience to walk to a bus stop, which is determined by the distance from the nearest bus stops. In China, the planning standard for urban comprehensive transportation systems pointed out that residents were more inclined to accept a walking distance of 5 min to 10 min, and this is used as a benchmark to propose the requirements of 300 m and 500 m in service coverage lands for public bus stop. Based on this requirement, this study sets 300 m as the range of bus stops with the best service radius, 300–500 m as the effective service radius, and 800 m as the maximum service radius of bus stops.

Functional diversity can reflect the diversity of land utilization. It is calculated using information entropy based on facility types within an 800 m buffer area surrounding the residential area. Facility types include restaurants, enterprises, leisure, schools, hospitals, the government, malls, and public spaces.

Road connectivity reflects the spatial relations of the road network surrounding the residential area. Space syntax is used to describe these spatial relations in the study by calculating three indicators, including global integration, control values, and connectivity metrics. The estimated range is an 800 m buffer area surrounding the respondent’s residential area.

The distance to the city center (and commercial center) refers to the distance between where the respondent lives and the landmark buildings in the city center (and commercial center). It reflects the regionality of where the resident lives and has been widely proven to be a significant factor in travel.

### 3.4. Study Case and Data

Zhengzhou City, Henan Province, is an important city in the central region of China and is a prominent nationally integrated transportation hub with a resident population of about 6.5 million in its central urban area. Due to urban areas with flat terrain, clear road networks, and high building and population densities, bus routes are widely covered within the urban area, and bus transport is a significant transport option for Zhengzhou citizens during their daily journeys. Therefore, residents within the central urban area of Zhengzhou City were targeted for our survey on the subjective psychological factors of bus travel intention, while information regarding the geographical location of the respondents’ residential areas was collected.

The questionnaire survey on subjective psychological factors was divided into two parts: a pre-survey and a formal survey. The pre-survey was conducted in a minor area for test collection, and the reasonableness of the questionnaire items was verified based on the survey’s results; the distribution of items in the questionnaire was adjusted to obtain the formal questionnaire. The formal survey was conducted in December 2021 at large shopping malls, parks, and other high-traffic public areas within all administrative districts of Zhengzhou to ensure a uniform spatial distribution of the location of the respondents’ settlements. We used the paid survey approach and used cash and souvenirs as incentives. Five hundred questionnaires were collected, and four hundred and ten valid questionnaires were returned, excluding those with incomplete answers, respondents who failed to check questions, and responses with unclear geographical information.

The data of the bus travel environment comprised the vectorial geographic information data, which came from Open Street Map and Gaode Map LBS open data, including POI, vectorial road network, bus routes, etc. Based on the location information of the residences collected from the survey, 410 survey samples covered 295 residential communities. The bus travel environment within walking distance of each residence was extracted and calculated, and the obtained data were normalized and then entered into the SEM model for calculation together with the subjective psychological questionnaire data. The district’s distribution and the personal attributes of the respondents are shown in Figure 3 and Table 3.

## 4. Empirical Analysis

### 4.1. Reliability and Validity Test

The results of the data reliability and validity tests are shown in Table 4. The reliability of the latent variables in this model was tested by Cronbach’s alpha, and the alpha coefficient was more than 0.7 for all except the perceptual behavior control, which was 0.691, indicating that there was better consistency with respect to all the data. All factor loading coefficients in the exploratory factor analysis (EFA) were higher than 0.5, indicating that the measured variables were well-expressed in terms of information, with clear ideographs for the question items and higher differentiation. The factor loadings in the validated factor analysis (CFA) was higher than 0.5 except for the SN3 variable of 0.488, and the average variance extracted values (AVE) were above 0.4, indicating that the measurement model had better explanatory power, and the latent variable settings had better aggregation [39,40,41,42].

### 4.2. Influence Path Testing of the Impact of Latent Variables Based on the SEM

The results of the model fit test are shown in Table 5. The model’s test metrics are as follows: the root mean square error of approximation (RMSEA), the goodness of fit index (GFI), comparative fit index (CFI), incremental fit index (IFI), and adjusted goodness of fit index (AGFI). According to the suggested values of each indicator, all tested indicators qualified for the model’s fit, indicating that the model had robust explanatory power. The calculated standardized path coefficients of each latent variable are shown in Table 6. The effects of all the psychological factors and latent variables on the intention to travel by bus (H1–H3) showed robust statistical significance. In terms of the path of influence of the travel environment’s latent variables, their relationships were validated (confidence interval 95%) for the intention to travel (H7) as well as attitude (H4). However, the relationship effects for the subjective norm (H5) and perceptual behavior control (H6) showed relatively less statistical significance (90% confidence interval). The latent variable path hypothesis was largely valid, and the final validated latent variable influence paths were obtained, as shown in Figure 4.

Factor analysis was performed on all the measured variables in SEM, and the fitness values of the latent variables were calculated as variables influencing behavioral intention based on their factor scores. The regression analysis was conducted using the value of the latent variable of bus travel intention as the regression outcome variable.

### 4.3. The Analysis of Bus Travel Intention Impact Factors

This study performed Pearson correlation tests on variable values, as shown in Table 7. Both travel environment variables and psychological factors showed significant positive correlations. The correlation coefficients between each variable were less than the critical value of 0.4, and it was concluded that there was no significant multicollinearity between the latent variables [43]. Thus, it was appropriate to use a linear regression model for analysis.

The influences of bus travel intentions are divided into three categories: psychological factors, travel environment, and personal attributes. The coefficient and validity are compared by controlling variables to determine the model. Three linear regression models were set up for the study, and then the models were compared with each other. Model A contained only psychological factor variables to reflect the influence of psychological factors on transit traveling intentions; Model B contained psychological factors and travel environment variables to reflect the interaction of psychological factors and travel environments relative to travel intention; Model C added personal attribute variables of travelers based on Model B to compare the differences in transit traveling intentions between different groups of personal attributes.

The linear regression model results are shown in Table 8. VIF is the variance inflation factor, and the VIF value of each indicator in the model is close to 1 [44,45]; moreover, there is no serious problem of multicollinearity. All variables in Model A and Model B showed statistical significance at 99% confidence intervals, indicating that both psychological and travel environment factors in SEM showed significant effects on transit traveling intentions. The retirement attribute in Model C is statistically significant, and all other personal attributes are not statistically significant, indicating that retired people are more inclined toward traveling by transit. The other attributes do not have significant characteristic differences in their bus travel intentions.

## 5. Results and Discussion

Urban regeneration with the aim of promoting a transformation in the way people travel is effective in reducing motor vehicle travel and environmental pollution. Many studies have been conducted to explain in detail the mechanisms of behavioral influence based on psychological factors using the TPB model. However, these studies have lacked the support of objective data, and the role of environmental and psychological factors combined with the behavioral intention to travel is not sufficiently considered. To address this issue, this study introduced the SOR theory based on the TPB theory to formulate a path hypothesis, which verified the influence path of environmental–behavioral intention on the one hand and the influence path of environmental–psychological factors as well on the other.

The study first proposed hypotheses H1, H2, and H3 based on the TPB theory and validated the psychological path of the TPB theory, which was evident and widely confirmed [16,46,47]. As with many previous studies based on the TPB theory, the causal link between the environment and behavior has never been established, although some valid conclusions have been drawn. Afterward, hypotheses H4, H5, H6, and H7 were formulated based on the SOR theory for the travel environment to determine behavioral intention. This conducted study showed that travel environment factors significantly affected the attitude factors and behavioral intention factors and simultaneously showed borderline significant results for subjective normative factors and perceived behavioral control factors. Therefore, the hypothesis of the “travel environment-psychological factors-behavioral intention” path has been valid. The significant association between the travel environment and traveling behavior has been widely demonstrated in many studies [48,49]. Therefore, this study’s results have been confirmed to some extent by other scholars’ studies.

The path relationship shows that the bus travel environment is used as an exogenous latent variable and has a significant effect on both the endogenous latent variables of psychological factors as well as a moderating effect on bus travel intention. The findings show that TE is an important influence on ATT and that ATT can further affect BI, which indicates that a good travel environment can increase people’s positive evaluation of bus travel. For example, increasing the frequency of bus departures and improving bus travel guidance signs can help people to use bus services more conveniently, thus increasing their bus travel intentions. In addition, the TE can also affect BI via the SN of travelers, which means that a better travel environment can increase the perceived social pressure for people to choose bus travel, thus enhancing the possibility of choosing bus travel. The impact of TE on the PBC indicates that a good bus travel environment enhances people’s perception of bus travel and has a positive effect on the eventual formation of bus travel decisions. The association between the built environment and bus travel is further reinforced by the direct effect of the built environment on bus travel intentions, confirming that the built environment not only influences bus travel decisions via subjective psychological factors but also acts as a constraint and control variable in the formation of bus travel intentions.

The path coefficients show that of the three psychological factor latent variables, PBC has the largest effect on BI, followed by ATT and SN. This suggests that improving the quality of the travel environment and reducing the difficulties faced by travelers when taking the bus play an important role in improving bus travel intentions. The TE influences BI in both direct and indirect ways, with distance the to city center (TE5) and functional diversity (TE2) having factor loadings that are above 0.8 for this latent variable measure; they are the main influencing factors for the travel environment latent variable. Figure 2 shows that the building density and building diversity in the central city of Zhengzhou is significantly higher than in the suburban areas, and when combined with the high correlation between the TE5 and TE2 variables, it can be concluded that Zhengzhou has obvious monocentric characteristics. From the perspective of urban planning, the gradual formation of a multi-center, multi-group urban spatial form through the establishment of urban sub-centers will help increase residents’ willingness to travel by public transport.

## 6. Conclusions and Prospects

Based on the combination of subjective and objective perspectives, this article attempts to analyze the influence of bus travel environments on public transport travel intentions by using subjective psychological factors as mediating variables. The main contributions of this study are as follows: (1) It confirms that psychological factors play a mediating role in the influence of built environments on travel intention; (2) It verifies the influence path of “built environment-psychological factors-behavioral intention”. (3) It establishes a feedback mechanism for the optimization of built environments.

This study verifies the mechanism of the influence of subjective and objective factors on travel intention and provides some suggestions for improving people’s choice of public transport. Due to the limitations of the authors, there are still some elements that have not been fully considered and dealt with in this study, such as the lack of subgroups of the characteristic population, the failure to consider the travel patterns with respect to purpose, and the need to further expand and refine the quantitative dimensions of the built environment.

## Figures and Tables

**Figure 1 behavsci-12-00462-f001:**
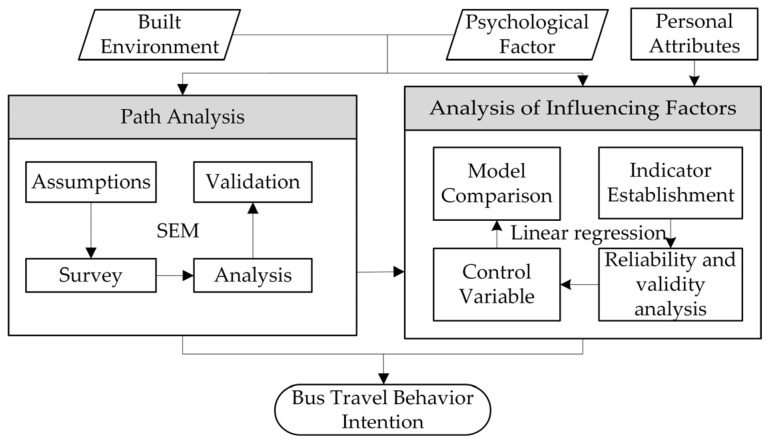
Research framework.

**Figure 2 behavsci-12-00462-f002:**
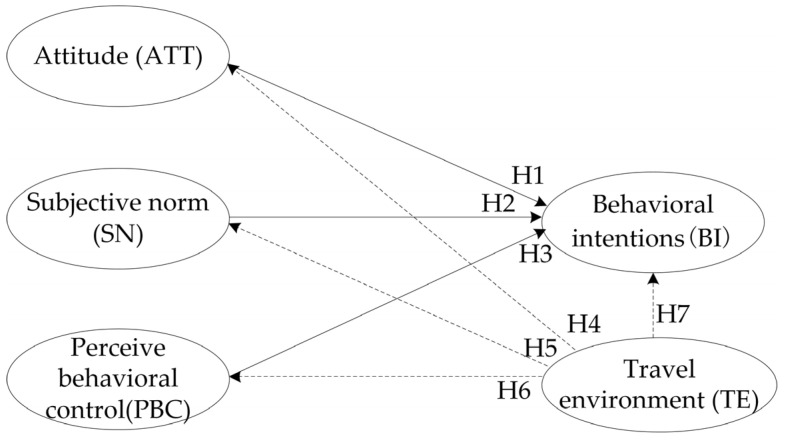
Structural model.

**Figure 3 behavsci-12-00462-f003:**
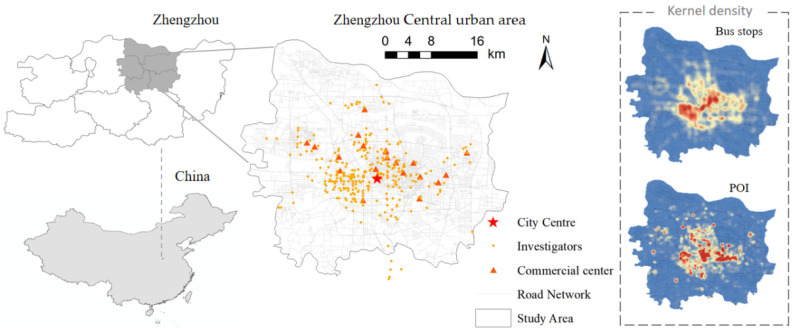
Research area and sample distribution.

**Figure 4 behavsci-12-00462-f004:**
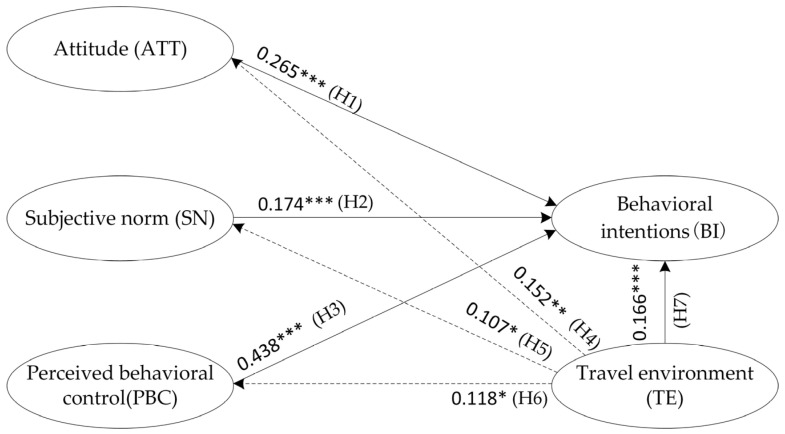
Result of structural equation model. Note: *** *p* < 0.01, ** *p* < 0.05, * *p* < 0.1.

**Table 1 behavsci-12-00462-t001:** Description of measurement variable.

Latent Variable	Measurement Variable	Serial	Latent Variable
ATT	Bus travel speed could satisfy my requirements	ATT1	An evaluation of whether a behavior is positive or negative.
	Bus travel could reduce environmental pollution	ATT2	
	Bus travel could reduce traffic congestion	ATT3	
SN	I select bus travel to alleviate urban pollution	SN1	Societal stress is derived from doing a certain act.
	I select bus travel to relieve city traffic congestion	SN2	
	My family and friends recommend bus travel	SN3	
PBC	I am acceptable of the waiting time when taking the bus	PBC1	Level of difficulty in completing an action.
	I can find bus stops very easily	PBC2	
	I do not feel crowded when I ride the bus	PBC3	
TE	Bus-stop accessibility	TE1	Quantification of the travel environment when choosing to travel by bus.
	Functional diversity	TE2	
	Road connectivity	TE3	
	Distance to commercial center	TE4	
	Distance to city center	TE5	
BI	Bus travel is my primary mode of commuting	BI1	Traveling intention of respondents.
	I prefer bus travel to other modes of travel	BI2	
	I would recommend my family and friends travel by bus	BI3	

**Table 2 behavsci-12-00462-t002:** Definition of the bus travel environment.

Measurement Variable	Schematic Diagram	Definition	Calculation Method
Bus-stop accessibility	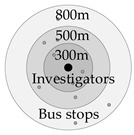	The more bus stops and the closer walking distance around the residence, the higher bus-stop accessibility.	$w=\Sigma w_{i}$ $l\leq 300\;m,\;w_{i}=1$ $300<l\leq 500\;m,\;0.5\leq w_{i}<1$ $500<l\leq 800\;m,\;0\leq w_{i}<0.5$
Functional diversity	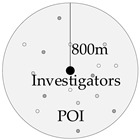	Information entropy was used to measure functional diversity. The functions of the buildings were classified accordingly into eight types: dining, enterprise, leisure, school, hospital, government, skyscraper, and shopping. Information entropy was then used to calculate the level of functional diversity in the area.	$E_{i}=-ln{\left(n\right)}^{-1}\overset{n}{\underset{j=1}{\sum }}p_{ij}lnp_{ij}$ $Q_{i}=\frac{E_{i}}{k\times 1-\sum_{i=1}^{n}E_{i}}$
Road connectivity	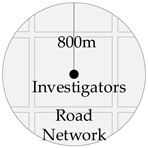	The three indexes involving global integration degree, control value, and the connectivity degree of the road network within the Zhengzhou city area are calculated by the spatial sentence method, and the road connectivity degree within the cell area is calculated by using information entropy.	
Distance to commercial center	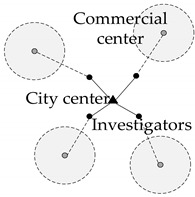	The spatial linear distance of the residential area is from the center of the nearest commercial area.	Measured by geographic information tools (ArcGIS)
Distance to city center		The spatial linear distance of the residential area from the city center.	

Note: Min-max standardization is conducted to obtain variable values. The distance to the business center and the distance to the city center are two variables that are expected to have negative impacts on bus travel intentions. The negative indicators need to be unified to the same direction as the other indicators through consistency processing.

**Table 3 behavsci-12-00462-t003:** Statistical description of personal attributes.

Personal Attributes	Group	Number of Samples	Ratio
Gender	Male	200	48.78%
	Female	210	51.22%
Age	≤18	10	2.44%
	18–30	254	61.95%
	30–40	81	19.76%
	40–55	36	8.78%
	55–65	12	2.93%
	65 and above	17	4.15%
Occupation	White-collar worker	38	9.27%
	Commerce and service	139	33.90%
	Blue-collar worker	98	23.90%
	Unemployed	28	6.83%
	Student	55	13.41%
	Retired	52	12.68%
Education	Primary school	2	0.49%
	Junior high school	20	4.88%
	Senior high school	58	14.15%
	Bachelor	284	69.27%
	Master and doctor	46	11.22%
Personal annual income (The monetary unit is CNY. The average level in 2022 is 96,400)	Within 50,000	189	46.10%
	50,000–100,000	125	30.49%
	100,000–150,000	57	13.90%
	150,000–250,000	19	4.63%
	200,000–250,000	7	1.71%
	250,000 and above	13	3.17%

**Table 4 behavsci-12-00462-t004:** Reliability and validity test.

Latent Variable	Measurement Variable	Alpha	EFA	CFA	AVE	Factor Score
ATT	ATT1	0.748	0.647	0.590	0.532	0.344
	ATT2		0.796	0.764		0.456
	ATT3		0.847	0.814		0.510
SN	SN1	0.772	0.885	0.862	0.610	0.479
	SN2		0.901	0.921		0.488
	SN3		0.544	0.488		0.254
PBC	PBC1	0.691	0.780	0.706	0.436	0.469
	PBC2		0.692	0.572		0.427
	PBC3		0.761	0.695		0.470
BI	BI1	0.780	0.742	0.640	0.534	0.410
	BI2		0.799	0.724		0.444
	BI3		0.816	0.817		0.449
TE	TE1	0.815	0.695	0.621	0.508	0.236
	TE2		0.848	0.803		0.292
	TE3		0.736	0.651		0.258
	TE4		0.659	0.574		0.227
	TE5		0.874	0.870		0.295

**Table 5 behavsci-12-00462-t005:** Test for goodness of fit.

Test Indicator	RMSEA	CFI	GFI	AGFI	IFI
Reference value	<0.08	>0.850	>0.850	>0.850	>0.850
Actual value	0.079	0.890	0.898	0.861	0.891

**Table 6 behavsci-12-00462-t006:** Path coefficients and hypothesis verification.

Path Coefficient	Path Coefficient	Hypothesis Verification
AT—>BI	0.265 ***	H1 is valid
SN—>BI	0.174 ***	H2 is valid
PBC—>BI	0.438 ***	H3 is valid
TE—>AT	0.152 **	H4 is valid
TE—>SN	0.107 *	H5 is valid
TE—>PBC	0.118 *	H6 is valid
TE—>BI	0.166 ***	H7 is valid

Note: *** *p* < 0.01, ** *p* < 0.05, * *p* < 0.1.

**Table 7 behavsci-12-00462-t007:** Pearson correlation test.

	ATT	SN	PBC	TE
ATT	1.000	—	—	—
SN	0.397 ***	1.000	—	—
PBC	0.409 ***	0.308 ***	1.000	—
TE	0.110 **	0.110 **	0.093 *	1.000

Note: *** *p* < 0.01, ** *p* < 0.05, * *p* < 0.1.

**Table 8 behavsci-12-00462-t008:** Verification of effective influencing factors.

Category	Explaining Variable	Model A		Model B		Model C	
		Coefficient	VIF	Coefficient	VIF	Coefficient	VIF
Psychological factors	ATT	0.205 ***	1.330	0.197 ***	1.334	0.207 ***	1.360
	SN	0.206 ***	1.224	0.197 ***	1.229	0.189 ***	1.242
	PBC	0.294 ***	1.238	0.288 ***	1.241	0.259 ***	1.292
Travel Environment	TE			0.132 ***	1.019	0.133 ***	1.111
Personal Attributes	Age					−0.048	1.428
	Education					−0.034	1.140
	Personal annual income					−0.031	1.326
	Male					0.037	1.071
	Awaiting employment					0.008	1.272
	Retirement					0.126 **	1.511

Note: *** *p* < 0.01, ** *p* < 0.05.

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
