# Peer review of "Impact of Subjective and Objective Factors on Bus Travel Intention"

_behavsci, 2022, doi:10.3390/bs12110462_

Round 1

Reviewer 1 Report

In this paper, the authors investigate the effects of some factors on bus travel behavior intention. Although this topic is interesting, the paper is poorly written. The language is poor, and the logic in the paper is not clear. It is hard for me to understand the paper well.

1.       Please polish the language to make the paper more readable.

2.       The motivation for investigating bus travel behavior intention is too weak. Also, the paper lacks a well-organized background on sustainable transportation. Please reorganize the introduction to highlight the background and importance of studying bus travel behavior intention. A good example of the advantages of using the bus can be found in Han et al., (2021).

3.       The last paragraph in the introduction provides the structure of the paper. But it is not well written. May the author can write “Section 2 reviews the related literature. Section 3….”

4.       The literature review part is messy and does not use a consistent form of expression. I did not find differences between this paper and previous studies. The literature review lists and summarizes some papers. However, what are the limitations of previous studies? What are the differences between your study and previous studies? The authors should highlight the difference between their study and previous studies in literature review. Also, when there is one author in a paper, it can be written Surname (Year). If there are two authors in a paper, it can be written Surname 1 and Surname 2 (Year). If the number of authors is greater or equal to three, it can be written Surname 1 et. al., (Year). I strongly recommend the authors reorganize the literature review section and highlight the differences between this paper and previous studies.

5.       In line 104, what is TAM? The authors should give the full words before an abbreviation.

6.       In lines 140-141. “(referring to Error! Reference source not found.)”. I suggest the authors carefully read the paper to correct the “typos.”

7.       The authors propose seven hypotheses. However, what are the theoretical backgrounds for these hypotheses? The authors should propose hypotheses based on previous studies. The authors can refer Du et al., (2021).

8.       Policy implications based on the results should be highlighted in the discussion.

References

Han, X., Yu, Y., Jia, B., Gao, Z. Y., Jiang, R., & Zhang, H. M. (2021). Coordination Behavior in Mode Choice: Laboratory Study of Equilibrium Transformation and Selection. Production and Operations Management, 30(10), 3635-3656.

Du, H., Zhu, G., Zheng, J., 2021. Why travelers trust and accept self-driving cars: an empirical study. Travel Behaviour and Society, 22, 1–9.

Author Response

Dear Reviewers
Our response is in the attached file, please see the attachment.

Reviewer 2 Report

Dear Authors,

thank your for the opportunity to review your manuscript. The research is delivered and described in an adeguate way. What I struggle to see is the critical approach to all these concepts, theory (planned behaviour) and methods. Do we need a regression model to say that pedestrians walk when they have enough place to do so?

The measurement variables are badly translated into English, correct them. What is "bus walkability"? I see the explanation, but I am not sure this term will be used by other researchers because it is confusing.

The authors want to study whether people take to bus or not in a certain city. They find that old people travel by bus. The authors try to explain this result by using concepts and theories that have been used circa 4.500.000 times in academia but have not much to do with Chinese retieree people on the bus.

In other papers, it has been discussed that the Theory of planned behaviour does not explain behaviour (for example, I plan to do a diate. This does not mean that I am on a diate nor that I will start eating less from tomorrow. it means that I am aware that I should do it.) The authors did not reflect at all on the concepts or theories they used in the study. They applied it and that's all. Calculations are done, SEM was carried out, but to study what? What is the purpose of this study? The authors gave some money to research participants, if they had given this money with a choice experiment, that would have made sense. I mean: you are sitting in a car, I give you 5 dollars to take the bus, will you take it?

Author Response

(The authors gave the same response as above.)

Round 2

Reviewer 1 Report

The authors have responded to my comments, and I recommend the paper be published in this form. I have only one minor comment. The last paragraph in the introduction provides the structure of the paper. Although the authors have rewritten the last paragraph according to my comment, it is wrong numbering. It should be "Section 2 reviews the related literature. Section 3...." Please carefully correct it.

Author Response

Dear Reviewers

Thanks to your suggestions that enabled us to avoid these errors. We have corrected them.

Line 69-72